# Providing new insights on the biphasic lifestyle of the predatory bacterium *Bdellovibrio bacteriovorus* through genome-scale metabolic modeling

**Cristina Herencias**[1¤], **Sergio Salgado-Briegas**[1,2], **M. Auxiliadora Prieto**[1,2], **Juan Nogales**[2,3]*

**1** Microbial and Plant Biotechnology Department, Biological Research Center-Margarita Salas, CSIC, Madrid, Spain, **2** Interdisciplinary Platform for Sustainable Plastics towards a Circular Economy-Spanish National Research Council (SusPlast-CSIC), Madrid, Spain, **3** Department of Systems Biology, Centro Nacional de Biotecnología, CSIC, Madrid, Spain

¤ Current Address: Department of Microbiology, Hospital Universitario Ramon y Cajal, Instituto Ramon y Cajal de Investigacion Sanitaria (IRYCIS), Madrid, Spain.
* j.nogales@csic.es

**Data Availability Statement:** All relevant data are within the manuscript and its Supporting Information files.

## Abstract

In this study we analyze the growth-phase dependent metabolic states of *Bdellovibrio bacteriovorus* by constructing a fully compartmented, mass and charge-balanced genome-scale metabolic model of this predatory bacterium (*i*CH457). Considering the differences between life cycle phases driving the growth of this predator, growth-phase condition-specific models have been generated allowing the systematic study of its metabolic capabilities. Using these computational tools, we have been able to analyze, from a system level, the dynamic metabolism of the predatory bacteria as the life cycle progresses. We provide computational evidences supporting potential axenic growth of *B. bacteriovorus*'s in a rich medium based on its encoded metabolic capabilities. Our systems-level analysis confirms the presence of "energy-saving" mechanisms in this predator as well as an abrupt metabolic shift between the attack and intraperiplasmic growth phases. Our results strongly suggest that predatory bacteria's metabolic networks have low robustness, likely hampering their ability to tackle drastic environmental fluctuations, thus being confined to stable and predictable habitats. Overall, we present here a valuable computational testbed based on predatory bacteria activity for rational design of novel and controlled biocatalysts in biotechnological/clinical applications.

## Author summary

Bacterial predation is an interspecific relationship widely extended in nature. Among other predators, *Bdellovibrio* and like organism (BALOs) have received recently a great attention due to the high potential applications such as biocontrol agent in medicine, agriculture, aquaculture and water treatment. Despite the increasing interest in this predatory

**Funding:** This work was supported by the Spanish Ministry of Economy and Competitiveness through the grants BIO2014-59528-JIN (JN), BIO2013-44878-R, and BIO2017-83448-R and H2020 Engicoin no 760994 (MAP). CH was supported by a FPI (Contratos predoctorales para la formación de doctores) fellowship (BES-2014-070856). SSB was supported by a FPU (Ayuda para la formación de profesorado universitario) fellowship (FPU17/03978) from the Spanish Ministry of Universities. The funders had no role in study design, data collection and analysis, decision to publish, or preparation of the manuscript.

**Competing interests:** The authors have declared that no competing interests exist.

bacterium, its complex lifestyle and growth conditions hamper the full exploitation of their biotechnological properties. In order to overcome these important shortcomings, we provide here the first genome-scale model of a predatory bacterium constructed so far. By using the model as a computational testbed, we provide solid evidences of the metabolic autonomy of this interesting bacterium in terms of growth, as well as its dynamic metabolism powering the predatory biphasic life style. We found a low metabolic robustness thus suggesting that *Bdellovibrio* is more niche-specific than previously thought and that the environmental conditions governing predation may be relatively uniform. Overall, we provide here a valuable computational tool largely facilitating rational design of novel and controlled predator-based biocatalysts in biotechnological/clinical applications.

## Introduction

Predation is a biological interaction where an individual, the predator, feeds on another, the prey, to survive. Since predation has played a central role in the diversification and organization of life, this system provides an interesting biological model from both an ecological and evolutionary point of view. Predation is an example of coevolution where the predator and prey promote reciprocal evolutionary responses to counteract the adaptation of each other [1]. This interspecific relationship is widely extended in nature, including the microbial world where the main predators are bacteriophages, protozoa and predatory bacteria [2]. Focusing on bacteria, this group is composed, among others, by *Bdellovibrio* and like organisms (BALOs) which are small, highly motile, and aerobic gram-negative predatory bacteria that prey on a wide variety of other gram-negative bacteria. Originally discovered in soil [3], BALOs are ubiquitous in nature. They can be found in terrestrial and aquatic habitats, bacterial biofilms, plants, roots, animals and human feces [4] and lung microbiota [5]. *B. bacteriovorus* is the best characterized member of the group of BALOs and the genome of different strains, including HD100, Tiberius and 109J have been sequenced providing a reliable source of genetic information [6–8].

*B. bacteriovorus* exhibits a biphasic growth cycle (Fig 1), including a free-swimming attack phase (AP) and an intraperiplasmic growth phase (GP) inside the prey´s periplasm forming the so-called bdelloplast structure. During AP, free living cells from extracellular environment are in active search for new preys. After attachment, and once the predator-prey interaction is stable and irreversible, the predator enters in the prey's periplasm, where it grows and replicates DNA during the GP using the cytoplasm of the prey cell as a source of nutrients and biomass building blocks. When the prey is exhausted, *B. bacteriovorus* grown as a filament, septates into several daughter cells, lyses the ghost-prey's outer cell membrane and releases into the medium [6,9]. Interestingly, host-independent (HI) mutants of *Bdellovibrio* strains have been found under laboratory conditions. These HI predators are able to grow axenically (without prey) in a rich-nutrient medium mimicking the dimorphic pattern of elongated growth, division and the development of the host-dependent (HD) cells following a multiple fission strategy [10]. It is worth noticing that the axenic growth of these mutant strains is given by a mutation in the host interaction (hit) locus, which has been described as being involved in regulatory and/or scaffold elements, such as type IV pilus formation and also related to the attachment and invasion of the prey [11]. This argues in favor of this mutation having no direct metabolic (enzymatic) impact. In fact, the main metabolism of these HI derivatives should not have suffered changes with respect to the wild type *Bdellovibrio* strains.

*B. bacteriovorus*' extraordinary repertoire of susceptible preys allows for a wide range of potential applications based on its predatory capability, such as biocontrol agent in medicine,

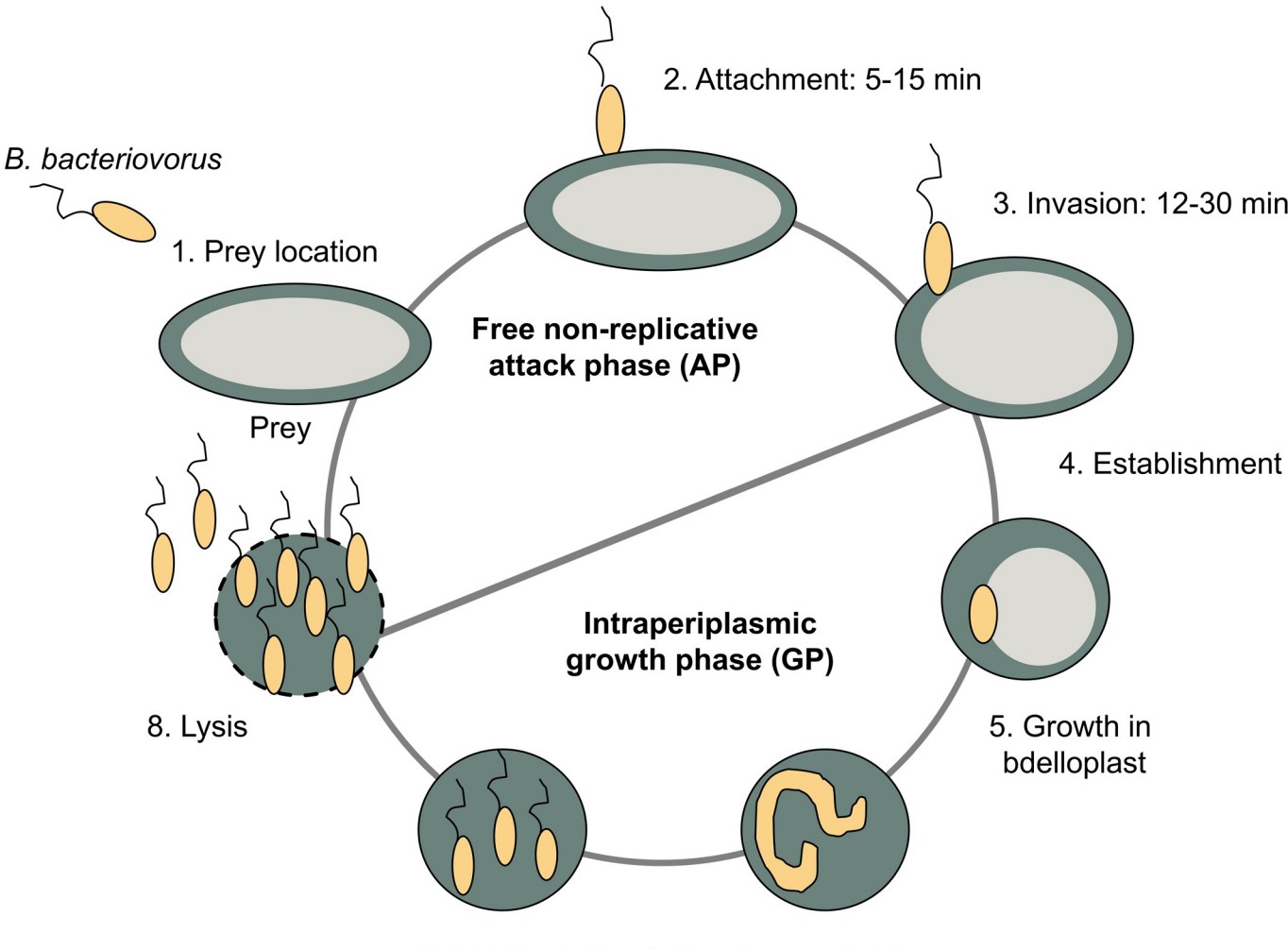

**Fig 1. Lifecycle of *B. bacteriovorus* HD100.** 1) Prey location: *B. bacteriovorus* moves towards prey-rich regions. 2) Attachment: the predator anchors to the host cell, which leads to the infection. 3) Invasion: *B. bacteriovorus* enters the periplasm of the prey cell. 4 and 5) Growth in bdelloplast and development: the prey has a rounded appearance due to cell wall modification and *B. bacteriovorus* grows in the periplasm and replicates its DNA. *B. bacteriovorus* uses the prey cytoplasm as a source of nutrients. 6 and 7) Septation and development: the predator septates when resources become limited and it matures into individual attack phase cells. 8) Lysis: mature attack-phase cells lyse the cell wall of the bdelloplast, initiating the search for fresh prey. The complete cycle takes about 4 h.

agriculture, aquaculture and water treatment [12–15]. Furthermore, it has been proposed as an excellent source of valuable biotechnological enzymes and as a biological lytic tool for intracellular products, due to its hydrolytic arsenal [4,16,17]. Moreover, regarding its unique lifestyle it represents a good model for evolution studies focusing, for example, on the origin of the eukaryotic cell [18,19]. Despite the interest that this predatory bacterium's potential applications have recently aroused among the scientific community, its complex lifestyle and growth conditions make it hard to implement metabolic and physiological studies. As a direct consequence, to date, its physiology and metabolic capabilities remain an enigma to a large extent [20]. Moreover, the potential of this predator to be used as a biotechnological chassis depends on the quantity and quality of the available metabolic knowledge. Therefore, expanding the knowledge of this predatory bacterium is essential for the full exploitation of its unique biotechnological applications. This process would require a reliable platform supporting the rational understanding of its characteristics.

Following this aim, the advent of genomic age and the subsequent large amount of derived high-throughput data, have largely contributed to deeper understanding of microbial behavior, at system level [21]. Specifically, genome-scale metabolic models (GEMs) are being used to analyze bacterial metabolism under different environmental conditions [22,23]. GEMs are structured representations of the metabolic capabilities of a target organism based on existing biochemical, genetic and phenotypic knowledge which can be used to predict phenotype from genotype [24].

The application of Constraint-Based Reconstruction and Analysis (COBRA) approaches [25] together with specific GEMs have been successfully applied for better understanding of interspecies interactions such as mutualism, competition and parasitism providing important insights into genotype-phenotype relationship [26]. Despite GEMs being powerful tools to elucidate the metabolic capabilities of single systems, addressing the complex metabolism of bacterial predators having biphasic growth-cycles such as *B. bacteriovorus* is challenging and has remained elusive so far.

We provide here the first step toward the metabolic understanding at system level of *B. bacteriovorus* by the reconstruction of its metabolism at genome-scale. We further use this cutting edge computational platform as a test bed for the integration and contextualization of transcriptomic and physiological data shedding light on the biphasic lifestyle of this predatory bacterium.

## Materials and Methods

### Genome-scale metabolic network reconstruction: *i*CH457

The genome-scale metabolic model of *B. bacterivorous* HD100 (*i*CH457) was constructed using standardized protocols for metabolic reconstruction [22,27], and is detailed in Fig 2A. An initial draft reconstruction was generated from the annotated genome of *B. bacteriovorus* HD100 (GenBank number: BX842601.2) using the automatic application provided by Model Seed server [28]. Additionally, the metabolic content of *B. bacteriovorus* was mapped with two broadly used and high-quality GEMs belonging to *E. coli* (*i*JO1366; [29]) and *P. putida* (*i*JN1411; [30]), generating additional drafts by using MrBac Server [31]. Once these models were unified into a final reconstruction, we proceeded to a thorough manual curation of the collected metabolic information. Genes proteins reactions (GPR) relationships was included following a Boolean logic to describe the nonlinear associations, where "and" corresponds to protein complex and "or" is related with isoenzymes. During this iterative process, the final inclusion of each individual's biochemical reaction was assessed using genomic [6], metabolic, transporter and GEMs databases, including: Kyoto Encyclopedia of Genes and Genomes (KEGG, [32]), BRENDA [33], BIGG [34]. Transport reactions were also added by using the TransportDB [35] database. Finally, we performed a manual gap filling step in order to connect the network as much as possible and remove potential inconsistencies.

The initial analysis reported incomplete biosynthetic pathways for some amino acids (e.g., glycine, serine, methionine, and tryptophan) and cofactors (e.g., thiamine, biotin). *B. bacteriovorus* legacy literature has been thoroughly consulted, ensuring high confidence in the metabolic content included. When specific data for the HD100 strain were not available, information from phylogenetically related organisms such as 109 Davis strain was used as previously suggested [27]. Relevant reactions added during this process were listed in S1 and S2 Tables. Charged and mass formulas for each metabolite, reaction directionality and stoichiometry, information for gene and reaction localization as well as gene-protein-reaction (GPR) associations for each reaction were carefully revised based on the available information for

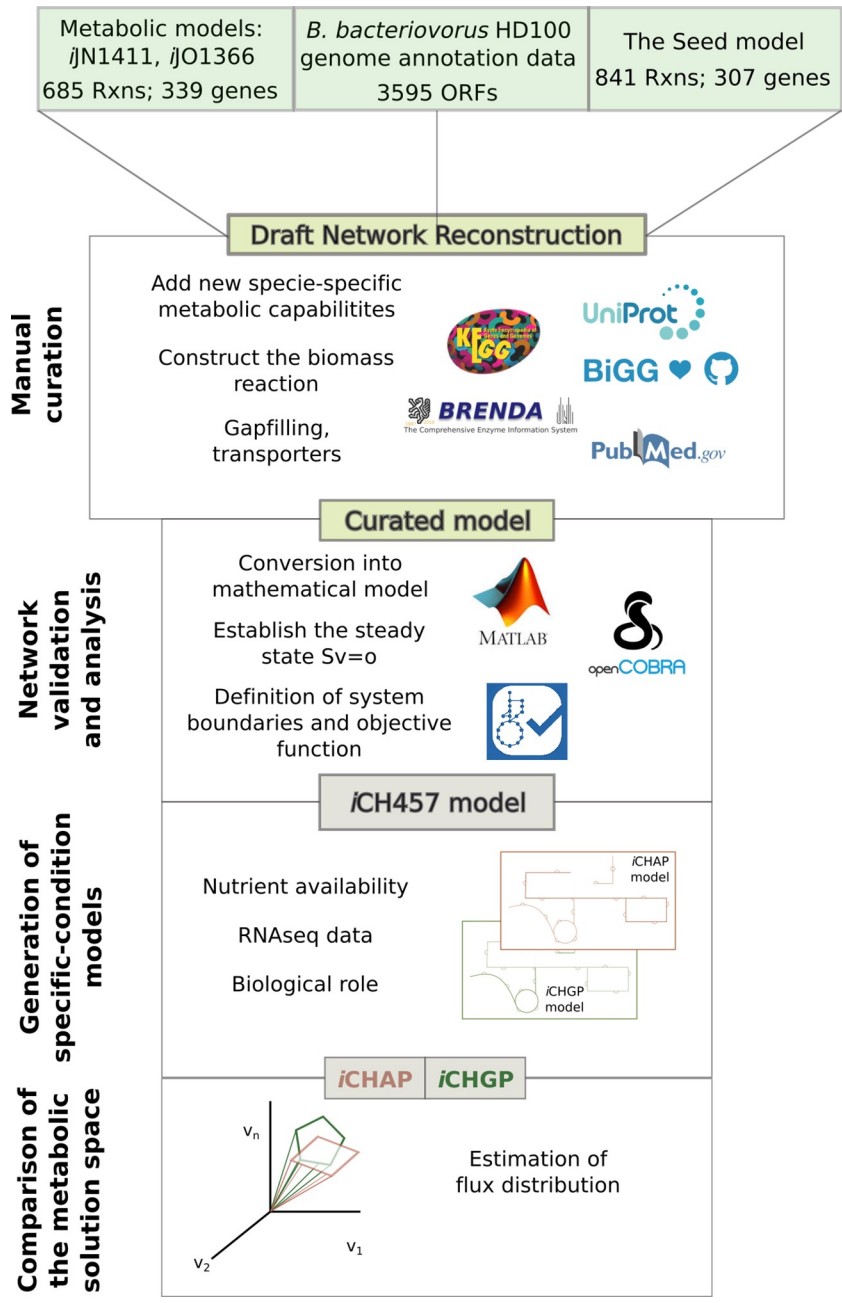

**Fig 2. *i*CH457 metabolic model pipeline.** The draft of metabolic reconstruction was based on available metabolic models (*i*JN1411 and *i*JO1366), the genome sequence of *B. bacteriovorus* HD100 and the automatic model Seed. Manual curation is required to accurately fine-tune the information contained in the metabolic model and several steps of network validation and analysis are required to finally obtain the metabolic model *i*CH457. The general model *i*CH457 was constrained based on nutrient availability (minimal and rich *in silico* media), biological role (ATP production or biomass generation) and transcriptomic available data [41] for the generation of condition-specific models: *i*CHAP and *i*CHGP. GIM3E algorithm was used to construct the condition-specific models Rxns: reactions.

*Bdellovibrio*. When this information was not available, data was collected from high-quality models.

The metabolites and reactions included in this metabolic model are listed in S5 and S6 Tables.

## Model analysis: Flux Balance Analysis (FBA)

FBA is by far the most popular approach for analyzing constraint-based models and it is used in many applications of GEMs. FBA uses optimization of an objective function to find a subset of optimal states in the large solution space of possible states that is shaped by the mass balance and capacity constraints. In FBA, the solution space is constrained by the statement of a steady-state, under which each internal metabolite is consumed at the same rate as it is produced [36].

The conversion into a mathematical format can be done automatically by parsing the stoichiometric coefficients from the network reaction list e.g. using the COBRA toolbox [37]. The dimensions of the stoichiometric matrix, **S**, are **m** by **n,** where **m** is the number of metabolites in the reaction network and **n** is the number of reactions. Therefore, each column represents a reaction and each row represents the stoichiometric participation of a specific metabolite in each of the reactions. FBA was used to predict growth and flux distributions. FBA is based on solving a linear optimization problem by maximizing or minimizing a given objective function to a set of constraints. The foundations and applications of FBA have been reviewed elsewhere [38,39]. A particular flux distribution of the network, **v**, indicates the flux levels through each of the reactions. Based on the principle of conservation of mass and the assumption of a steady state, the flux distribution through a reaction network can be characterized by the following equation: $S \times v = 0$ [36,40]. Constraints are placed on individual reactions to establish the upper and lower bounds on the range of flux values that each of the reactions can have. These constraints are described as follows: $\alpha_1 \leq v_1 \leq \beta_1$, where $\alpha 1$ is the lower bound on flux $v1$, and $\beta 1$ is the upper bound. If no information about flux levels is available, the value of $\alpha_1$ is set to zero for irreversible fluxes. In all other cases, $\alpha_1$ and $\beta_1$ are left unconstrained, thus allowing the flux to take on any value, whether positive or negative.

## Biomass function

It is commonly assumed that the objective of living organisms is to divide and proliferate. Thus, many metabolic network reconstructions have a so-called biomass function, in which all known metabolic precursors of cellular biomass are grouped (e.g. amino acids, nucleotides, phospholipids, vitamins, cofactors, energetic requirements, etc.). Since no detailed studies about *B. bacteriovorus* biomass composition are available, the biomass composition from *P. putida* [30] was used as a template for the biomass function of *i*CH457. However, data from *B. bacteriovorus* were added when available (e.g. nucleotide composition from genome sequence). Growth-associated ATP maintenance reaction (GAM), which represent the energy needed for cell replication and the non-GAM reaction (NGAM), related with the requirements to maintain other cellular functions was taken from the *E. coli* biomass reaction [29]. The detailed calculation of biomass composition is provided in S7 Table.

## Generation of growth phase-specific models: *i*CHAP and *i*CHGP

A given metabolic reconstruction is defined by the metabolic content contained in the genome and thus is unique for the target organism. However, it is possible to construct different condition-specific models by applying additional constraints such as condition-specific data (including physiological), gene/protein expression and flux data, etc.

To construct condition-specific metabolic models we incorporated these additional constraints to the model by means of a stepwise procedure including condition specific: i) biomass, ii) nutrient availability and iii) gene expression data (Fig 2B). Firstly, the objective function was adjusted to the biological role of AP and GP. ATP maintenance and biomass equations were selected as objective functions for AP and GP, respectively. In addition,

different *in silico* media were designed for each phase, simulating the availability of nutrients in each growth phase (S1 Text). Finally, available AP and GP gene expression datasets [41] were incorporated in order to constrain even further the solution space using GIM3E [42]. GIM3E builds reduced models by removing the reactions not available in the expression dataset while preserving model functionality. It should be noted that GIM3E considers which genes are expressed or not, but not the modifications in mRNA levels under different experimental conditions. A given gene was considered expressed when its RNA levels in the RNA-seq analysis fell within the first quartile, which is $\geq 10$ RPKM (Reads Per Kilobase Million) using the available dataset [41].

The distribution of possible fluxes in the specific-condition models was calculated using Markov chain Monte Carlo sampling [37]. This analysis is independent of objective function or other constrains taken into account previously to construct the specific condition models. The median value from the distribution was used as the reference flux value.

### Reactions essentiality analysis

In order to determine the effect of a single reaction deletion, all the reactions associated with each gene in *i*CH457 were individually suppressed from the matrix **S**. FBA was used to predict the mutation growth phenotype. The *singleReactionDeletion* function implemented in the COBRA Toolbox [37] was used to simulate knockouts. A lethal deletion was defined as that yielding < 10% of the original model's growth rate values. The simulations for reaction essentiality were performed using the rich *in silico* medium for *i*CH457 (S1 Text). Reaction essentiality analysis has been performed for other bacteria: *P. putida* KT2440 (*i*JN1411) [30], *E. coli* strain K-12 substrain MG1655 (*i*JO1366) [43], *Geobacter metallireducens* GS-15 (*i*AF987) [44], *Yersinia pestis* CO92 (*i*PC815) [45], *Salmonella* enterica subsp. enterica serovar Typhimurium str. LT2 (STM_v1_0 model) [34] and *Shigella flexneri* (*i*SF1195) [34].

The associations between essential reactions and each bacterium were represented building a bipartite network. For visualization we use Gephi software (0.9.2).

### Software

The *i*CH457 model was analyzed with the COBRA Toolbox v2.0 within the MATLAB environment (The MathWorks Inc.) [46]. Tomlab CPLEX and the GNU Linear Programming Kit (http://www.gnu.org/software/glpk) were used for solving the linear programing problems.

## Results

### Characteristics of *B. bacteriovorus* metabolic reconstruction

A genome-scale metabolic model (*i*CH457) including the metabolic content derived from genome annotation and available biochemical information was created for *B. bacteriovorus* HD100. *i*CH457 does not differentiate AP and GP, but it is a powerful tool for determining and analyzing the potential metabolic capabilities of the system from a global perspective. All the gene-protein-reaction associations (GPRs) included in the model were subject to a rigorous manual curation process in order to ensure the quality of the final model. Several open reading frames (ORFs) were annotated de novo and/or re-annotated during the reconstruction process. For instance, from the initial 75 ORFs included in the reconstruction draft belonging to amino acid metabolism based on bioinformatics evidence, only 65 (87%) were finally included according to bioinformatics and literature-based evidences. Moreover, during the manual curation process, we confirmed (by sequence homology and further metabolic contextualization the function of several genes related to amino acids metabolism and the hydrolytic

enzymes involved (S1 and S2 Tables). For instance, gene *bd0950*, annotated as an unspecific acetyltransferase, was specifically associated with an UDP 2,3-diamino-2,3-dideoxy-D-glucose acyltransferase, while gene *bd2095*, firstly annotated as encoding for an acetyl-CoA C-acetyl-transferase, was unequivocally re-annotated as a 3-ketoacyl-CoA thiolase. Similarly, *bd1852* which was initially annotated as an unspecific Enoyl-CoA hydratase in the genome of *Bdellovibrio*, the homology-based draft reconstruction identified this gene as encoding for one of the 3 methylglutaconyl CoA hydratases acting in the branched chain amino acids metabolism. Finally, manual gap-filling analysis and literature legacy suggested the participation of this gene as a putative Methylmalonyl-CoA decarboxylase [47].

*i*CH457 includes 457 ORFs, which represent 13% of the coding genes in the genome, whose gene products account for 705 metabolic and transport reactions (accounting for 70.5% of the model's total reactions). The model was completed with the inclusion of 296 non-gene associated reactions (29.5%) based on physiological and/or biochemical evidences supporting their presence in *B. bacteriovorus*. For instance, reactions related to the ACP acyltransferase (G3PAT) needed for glycerophospholipid biosynthesis were included based on the physiological evidence provided by Nguyen and col. and Muller and col. [47,48]. Overall, *i*CH457 accounts for a total of 1001 reactions and 955 metabolites distributed in three different compartments: cytoplasm, periplasm and extracellular space.

Reactions from *i*CH457 fall into 12 main functional categories (Fig 3). It is noteworthy that cell envelope metabolism seems to be the most represented group with a total of 222 reactions. In this important group we found reactions involved in the metabolism of peptidoglycans, lipopolysaccharides, glycerophospholipids, and murein. Across this group, catabolic reactions including reactions involved in the degradation of peptidoglycan by specific carboxypeptidases represent up to 37%. This high number of hydrolytic reactions present in *i*CH457 is consistent

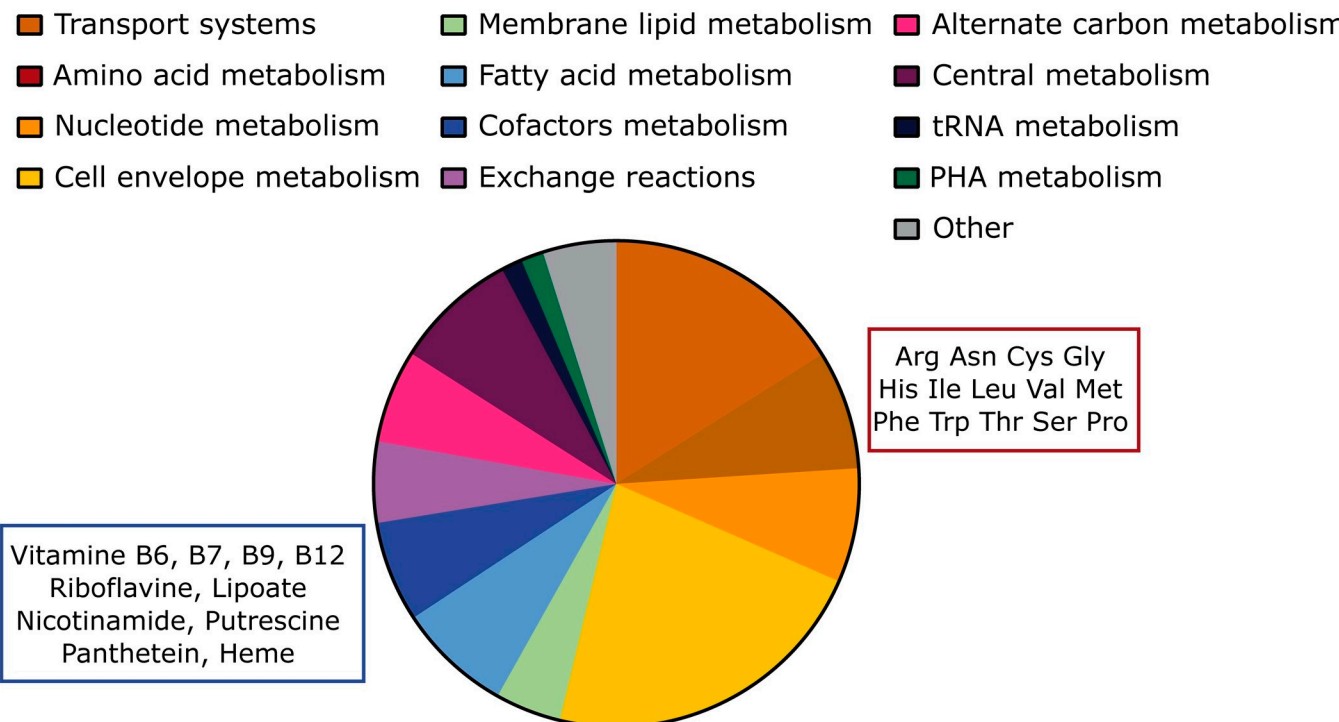

**Fig 3. Distribution of the reactions of the system in 12 global functional categories.** The metabolites inside the rectangles correspond with the auxotrophies in the cofactor metabolism (dark blue fraction) and the amino acid metabolism (brown fraction), respectively.

with the important role of these enzymes in the degradation of the prey's cell wall to penetrate into the periplasm, completion of the growth cycle and also in recycling the envelope components [49]. However, among the dead end reactions present in the model, those related to lipids and peptidoglycan metabolism are over represented, suggesting a still incomplete cell envelope biosynthesis in the model. Therefore, this subsystem supposes a clear target for model expansion in future studies.

In the past 15 years, GEMs have garnered considerable research attention and numerous metabolic reconstructions have been generated for several organisms [50]. The metabolic models within the δ-proteobacteria group are underrepresented among this phylum and only a few of them have been constructed, for instance for *Geobacter* spp. and *Desulfovibrio vulgaris* [51–54]. Thus, the model of *B. bacteriovorus* HD100, *i*CH457, represents a new model within this group, which, as depicted in Table 1, provides a complete reconstruction of this important bacterial group in terms of the metabolites and reactions included (confidence score = 2.1). Furthermore, other microbial interactions have been recently modeled, such as the syntrophic association between *Desulfovibrio vulgaris* Hildenborough and *Methanococcus maripaludis* S2 [52]. The metabolic reconstruction of this interaction highlighted the potential use of *in silico* predictions to capture growth parameters and community composition of bacterial communities. However, this is the first metabolic model of a predatory bacterium.

In order to ensure the quality of the model, we used the MEMOTE tool (https://memote.io/) [55] to evaluate the completeness and consistency of the metabolic network. The model's overall score was 88%, which suggests a good model completeness (SI2). The consistency of the model scored 99%, which represent the accuracy in stoichiometry, mass and charge balances, connectivity of metabolites and reaction cycles. This analysis confirms *i*CH457 as a complete and detailed model. The incomplete annotation in the genome of the predator can be related with some difficult to use the model with certain bioinformatics tools of script, but the accuracy and usability should not be affected.

## Model-driven assessment of auxotrophies and biomass building block transport systems highlight the predatory lifestyle of *B. bacteriovorus*

During the reconstruction process, we identified several incomplete biosynthetic pathways (lipids, amino acids, cofactors, and vitamins). Many of these metabolic gaps could be solved based on experimental evidences and when a specific gene could not be assigned, we fill the gaps by using orphan reactions (reaction not gene-associated). Examples of this include: the enoyl ACP reductase involved in the fatty acid biosynthesis as it is evident that lipids are synthetized in *Bdellovibrio* despite the gene encoding this activity is unknown. However, many other gaps, mainly involved in amino acid biosynthesis, could not be solved, concluding that these gaps indeed could be responsible of known and unknown autotrophies which is consistent with the numerous auxotrophies previously reported for strain HD100 [6]. Model-based

**Table 1. Comparison of the metabolic properties of *i*CH457 compared with other δ-proteobacteria (*Geobacter* spp. and *Desulfovibrio vulgaris*) metabolic models and with the well-establish metabolic reconstruction of *P. putida* (*i*JN1411) and *E. coli* (*i*JO1366).**

|  | *i*JN1411 | *i*JO1366 | *Geobacter metallireducens* | *Geobacter sulfirreducens* | *Desulfovibrio vulgaris* Hildenborough | *i*CH457 |
|---|---|---|---|---|---|---|
| **Protein-coding Genes** | 5350 | 4405 | 3532 | 3530 | 3379 | 3584 |
| **Genes (%)** | 1411 (26%) | 1366 (31%) | 747 (21%) | 588 (17%) | 744 (22%) | 457 (13%) |
| **Metabolites** | 2057 | 1136 | 769 | 541 | 1016 | 956 |
| **Reactions** | 2754 | 2251 | 697 | 529 | 951 | 1001 |
| **Transport Reactions (%)** | 650 (23%) | 592 (24.8%) | 62 (8.9%) | 50 (9.4%) | 93 (9.7%) | 181 (18%) |
| **Reference** | 30 | 43 | 51 | 54 | 53 | This work |

analyses provide an integrated overview of the complete metabolic network of this predator, including metabolic gaps potentially responsible for the auxotrophies. This is because instead of analyzing just the main synthetic pathways, such *in silico* analyses consider the global metabolism, including alternative and/or secondary/accessory biosynthetic routes.

In fact, model-based analyses identified up to 24 different auxotrophies. For instance, from the 20 proteinogenic amino acids, external supply of 14 of them was required to achieve *in silico* growth, including arginine, asparagine, cysteine, glycine, histidine, methionine, leucine, isoleucine, valine, phenylalanine, tryptophan, threonine, serine and proline. In addition, external supply of several cofactors including riboflavin, nicotinamide, putrescine, folic acid, pantothenate, pyridoxal phosphate, biotin and lipoate was needed in order to achieve *in silico* growth (Fig 3).

Concerning nucleosides monophosphate, we found that *B. bacteriovorus* has the ability to synthetize *de novo* all these key biomass building blocks despite nucleosides derived from the hydrolysis of prey having been traditionally suggested as a source of nucleic acids [56]. Supporting this computational analysis, radiotracer studies showed that strain 109J mainly utilized host nucleosides monophosphate during intraperiplasmic growth, however it was also able to synthesize its own pool of nucleotides [57,58]. This phenomenon has been traditionally explained in the context of an "energy-saving" mechanism. Evolution has promoted the loss of some genetic elements in bacteria that comprise cellular fitness and are usually related with the environment, when some metabolites are sufficiently present in the bacterial growth environment. For instance, amino acid biosynthetic pathway are sometimes partially represented as described in [59,60] where the organisms are saving the energy of biosynthetic costs.

Similarly, this mechanism has also been reported, and validated by our *in silico* analysis, for phospholipid assimilation and the recycling of some unaltered or altered fatty acids from the prey. Thus, while model analysis confirmed a complete and likely functional fatty acid *de novo* biosynthetic pathway, the direct uptake of these biomass building blocks has been largely reported. [47,61].

Due to their lifestyle and the obligate requirement of obtaining essential biomass building blocks from prey, the transport subsystem became an important key for the survival of *B. bacteriovorus*. In fact, this category was found to be one of the most representative in terms of number of reactions (181), highlighting their importance in cellular interchange compared with the transport reactions in other δ-proteobacteria metabolic models (Table 1). Although a comprehensive analysis of the transport systems in the predator has been previously reported [62], the predicted substrate specificity needs more experimental support. Remarkably, *i*CH457 model accounts for 67% of the annotated transport system reported in the genome.

It is worth emphasizing the case of peptide transporters; despite amino acids from protein breakage having been suggested as major carbon and energy sources during the intraperiplasmic growth of *B. bacteriovorus* [63], we noticed a significant lack of specific amino acid transporters during the model reconstruction and functional validation process. Instead, we found a large number of di- and tripeptide transporters, suggesting that the predator might be taking up small peptides from the prey.

Overall, model-based analyses largely supported the presence of energy-saving mechanisms in *B. bacteriovorus* targeting the biosynthesis of nucleotide monophosphates and phospholipids, but not of amino acids or vitamins whose availability depends exclusively on the prey. Likewise, detailed analysis of transport systems included in the model suggests *B. bacteriovorus*' ability to obtain oligopeptides through prey proteins cleavage and use them as its main source of carbon, nitrogen and energy during GP.

### *i*CH457 exhibits high accuracy predicting physiological states of *B. bacteriovorus* under different nutrients scenarios

A model's capability of providing accurate predictions of empirically-supported knowledge of a target organism's functional states is a key feature in order to assess the accuracy and completeness of the final reconstruction. However, the obligate predatory lifestyle of *B. bacteriovorus* and the complex environment provided by the prey, in terms of nutrients, prove challenging when using classical validation workflows based on single nutrient sources. Therefore, for *i*CH457 validation, we took advantage of spontaneous HI *Bdellovibrio* strains developed under laboratory conditions. Such HI strains exhibit an elongated cell development during the growth cycle in a rich medium which resemble the growing phase inside the intra-periplasmic space of the prey of the HD strains [64]. Indeed, because the HI phenotype has been attributed to putative regulatory and/or scaffold mechanisms rather than to metabolic genes (enzymes) [65,66], these HI strains are supposed to possess metabolic capabilities identical to those of the parental strains. Thus, for the GEMs validation process, including potential carbon sources and biomass generation rates, we decided to use data from HI *Bdellovibrio* strains for *i*CH457 validation.

Specifically, we validated the predictive capabilities of the *i*CH457 by comparing *in silico* results with experimentally determined biomass production and growth rates of the HI strain *B. bacteriovorus* 109 Davis [67]. The *in silico* growth rates were calculated using minimal medium supplemented with selected carbon sources (S1 Text). *i*CH457 was very precise predicting growth rate on five different carbon sources, since the analysis calculated with precision the results in the case of glutamate, glutamine and succinate, and for pyruvate and lactate the model predict the 70% of the actual value (Fig 4A). The slight discrepancies found between *in silico* predictions and *in vivo* results might be explained by an incomplete formulation of biomass function or higher energy maintenance requirements under the simulated conditions not accounted for in the current reconstruction. In addition, higher *in silico* growth rates are

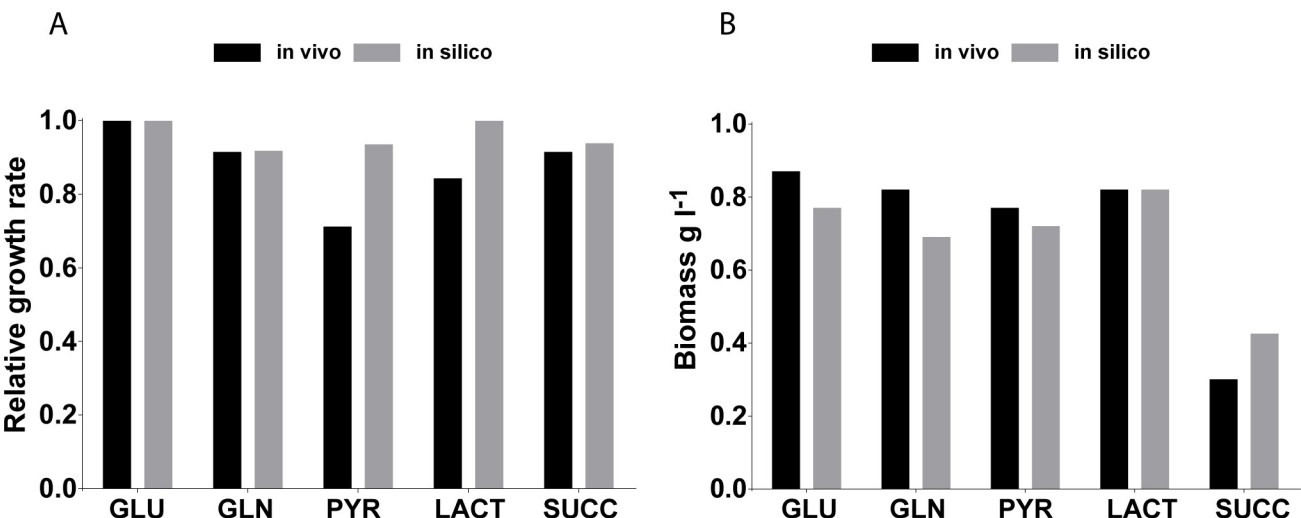

**Fig 4. Evaluation of the metabolic capabilities of *i*CH457. A) Comparison of the growth performance of the *in silico* *i*CH457 strain and a derivative strain of *B. bacteriovorus* 109 Davis on different carbon sources.** Experimental values of growth rate were calculated using the mass doubling time previously compiled in Ishiguro et al. [67]. The *in silico* growth rate was calculated with the minimal medium defined in S1 Text supplemented with the tested carbon source. B) Comparison of the biomass production predicted *in silico* with the available experimental data performed with the prey-independent *B. bacteriovorus* 109 Davis [67]. *In vivo* and *in silico* biomass data were expressed as Kendall's rank correlation coefficient ($\tau = 0{,}88$) for *i*CH457. GLU: glutamate, GLN: glutamine, PYR: pyruvate, LACT: lactate, SUCC: succinate. The statistically significant differences were calculated using two-way ANOVA followed by Bonferroni test. All comparisons were found non-significant ($P<0.05$).

often found due to the intrinsic nature and limitations of the flux balance analysis (FBA). FBA presumes a final evolution state in stark contrast with the potential scenario found *in vivo* which could lack the proper adaptation to these metabolites as a primary carbon source [68]. Also, FBA only predicts steady-state fluxes and does not account for any regulatory constraints, which should play an important role in the uptake of substrates from the extracellular medium [36]. Overall, our model predictions showed a significant accuracy that is comparable to other high-quality genome-scale models already available [30].

Beyond the availability to predict growth rates, it is valuable to assess the model's ability to predict the maximum amount of biomass produced from known concentrations of given carbon and energy sources. Similar high accuracy was found regarding the predictability of biomass production between *in silico* and experimental data (Kendallʻs coefficient τ = 0.88) (Fig 4B). It is noteworthy that the *in silico* analysis provided in these evaluations largely confirmed the prey-independent metabolic states, thus shedding light on the predator's potentially autonomous metabolism. These results are in good agreement with the large amount of HI derivative strains isolated previously [69] and the recent description of the metabolic response of AP cells in NB medium to synthesize and secrete proteases [70]. Therefore, the obligate predatory lifestyle of *B. bacteriovorus* should be questioned, at least from a metabolic point of view.

Overall, the high accuracy exhibited by *i*CH457 encourages us to use the model to characterize and better understand the metabolic states that underline the biphasic growth cycle of *B. bacteriovorus*.

## Reaction essentiality towards understanding the predator´s lifestyle

It is well-known that the environmental conditions and natural habitat of a given bacterium largely influence its evolutionary traits, including processes of genome expansion/reduction. Therefore, and taking advantage of *i*CH457, it would be interesting to address from a computational perspective whether the genome content of the predator has been influenced by its complex lifestyle. We identified a set of essential reactions in *i*CH457. The network reaction(s) associated with each gene was individually "deleted" by setting the flux to 0 and optimizing for the biomass function. A reaction was defined as essential if after constrained, the growth rate decreased to less than 10% of wild type model. To properly contextualize the reaction essentiality analysis, we compared our results with those from some free-living organisms such as *P. putida* KT2440 (*i*JN1411), *E. coli* strain K-12 MG1655 (*i*JO1366) and *Geobacter metallireducens* GS-15 (*i*AF987), as well as with other bacteria that also possess intracellular stages during their growth cycles, such as *Yersinia pestis* CO92 (*i*PC815), *Salmonella enterica* subsp. *enterica* serovar Typhimurium str. LT2 (STM_v1_0 model) and *Shigella flexneri* (*i*SF1195).

This reaction essentiality analysis showed no significant correlation between the number of essential reactions and the size of the metabolic network or the microorganism's lifestyle. The number of essential reactions found ranged from 214 to 419, with *Y. pestis* and *P. putida* being the organisms with lower and higher number of essential reactions, respectively. Moreover, the number of these essential reactions for the δ-proteobacteria, *B. bacteriovorus* and *G. metallireducens*, account for approximately 30% of the total reactions (Table 2 and S1 Fig). This rate could be related with the lack of a secondary metabolism in this bacterial group, which should be explored in depth in order to increase the computational value of the results.

The comparison of the essential reactions of the free living organism and the intracellular bacteria provided three main groups of essential reactions (exchange reactions were excluded from the comparison) (Fig 5): i) shared essential reactions between free-living and intracellular microorganisms (38 reactions), ii) free-living microorganisms' exclusive essential reactions (27 reactions), iii) intracellular microorganisms' exclusive essential reactions (15 reactions),

**Table 2. Reaction essentiality of the metabolic models of different bacteria.**

|  | *E. coli* | *P. putida* | *G. metallireducens* | *B. bacteriovorus* | *Salmonella* | *S. flexneri* | *Y. pestis* |
|---|---|---|---|---|---|---|---|
| **Total reactions** | 2583 | 2826 | 1285 | 1001 | 2545 | 2630 | 1961 |
| **Essential reactions**\* | 274 | 404 | 387 | 305 | 333 | 264 | 210 |
| **% of essential rxn** | 10.6 | 14.6 | 30.1 | 30.2 | 13.1 | 10.3 | 10.7 |

\* Exchange reactions were excluded.

related with cell envelop metabolism, glycerophospholipid metabolism, transport and nucleotide metabolism. Potentially, the 38 shared reactions would be part of the hypothetical essential metabolic network. Overall, the reactions found in the shared essentiality group are related with cell envelope, nucleotide, and cofactors (S3 Table).

Among the essential reactions found exclusively in the group of free-living microorganisms, none of them are present in the *i*CH457 metabolic model and they are mostly involved in cell envelope biosynthesis. This result together with the predator's auxotrophies suggest the adaptation of *B. bacteriovorus* to a non-free-living lifestyle, where the uptake of metabolites becomes crucial to its survival. Moreover, numerous reactions involved in amino acid metabolism are included in the group of free-living organisms, but not in the predator set, likely due to the direct incorporation of these metabolites from the prey.

### Analysis of the predator´s lifestyle using condition-specific models: Attack Phase (*i*CHAP) and Growth Phase (*i*CHGP) models

*B. bacteriovorus* possesses a biphasic growth cycle, leading by an extracellular attack phase (AP) and an intraperiplasmic growth phase (GP). It has been previously reported that these two stages are clearly differentiated in terms of gene expression [41] and also in the biological role [20], changes that must be strongly determined by the microenvironment.

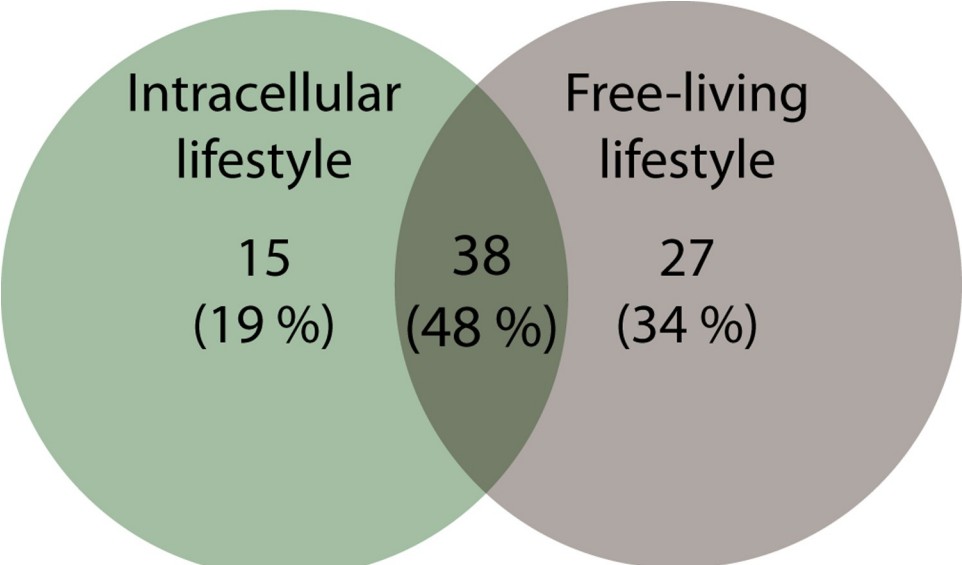

**Fig 5. Comparison of the reaction essentiality of intracellular lifestyle and free-living bacteria.** Numbers of reactions and the corresponded percentage are represented. Only reaction included within *i*CH457 were considered for comparison.

*B. bacteriovorus* AP cells are exposed to extracellular environment with highly diluted concentration of nutrients, but during GP the predator finds a very rich environment inside of the prey. As a consequence, it is reasonable to assume that the predator could hardly find nutrients during AP, which defines the search, attachment to and invasion of new preys as its main biological objective under this scenario. It could be envisaged that, during this period, the predator's metabolism is rerouted to obtaining energy, in terms of ATP, thus allowing flagellum movement and facilitating the collision with prey cells thanks to its high velocity [71]. Once the predator enters the prey periplasm it uses the cytoplasm as a source of nutrients to initiate GP. Bacterial cytoplasm is a very crowded compartment where most of the components of a microorganism are localized (30–40% of macromolecules and over 70% of proteins [72]). Thus, the cytoplasm is an extremely rich environment supporting growth and the completion of *B. bacteriovorus*' life cycle [73,74]. Consequently, it is reasonable to hypothesize that the main aim of this phase is to grow, which implies a highly active metabolism (catabolism and anabolism) supporting fast biomass generation. In fact, recent transcriptomic analyses have shown a highly activated anabolism in this phase [41].

To obtain a deeper understanding of this predatory bacterium in each phase of its life cycle, we constructed two different condition-specific models. Overall, these GP and AP-condition models were constructed by constraining *i*CH457 in terms of: i) nutrient availability, ii) biological objective, and iii) gene expression profile. As a first step, based on the environmental conditions, we defined two different *in silico* media: e.g., minimal and rich medium for AP and GP, respectively (S1 Text). Secondly, focusing on biological role, we used different biological objectives for simulating AP and GP phases. Thus, under AP and GP, ATP production and biomass production were selected as differential objective functions, respectively. Finally, in order to constrain even more the solution space in each model, data from RNA-seq analyses collected during AP and GP [41] were integrated into the metabolic model by using GIM3E [75]. GIM3E is an algorithm which minimizes the use of reactions whose encoding gene expression levels are under a certain threshold and finds a flux distribution consistent with the target function (biomass generation for GP or ATP production for AP). Following this workflow, we constructed two new models (*i*CHAP and *i*CHGP), mimicking AP and GP growth phases, respectively (Fig 2B).

The number of reactions of each specific-condition model was significantly reduced (from 1001 to 810 and 841 in AP and GP, respectively). This significant reduction involves reduced solution spaces, and thus likely more accurate predictions. As could be inferred given the difference in biological objectives in each phase, the condition-specific models were significantly different regarding the specific metabolic content (Fig 2B and S4 Table). For instance, while we found several reactions only present during AP (67 reactions), including reactions involved in glycerophospholipid degradation, the β-oxidation pathway and a large number of reactions were only present in GP model (98 reactions) i.e. reactions responsible for the biosynthesis of the cell envelope, nucleotides, fatty acids and lipids (S4 Table). In other words, we found that while the unique enzymes present during AP were mainly involved in energy production and cell survival, during GP the reactions were largely involved in anabolic pathways including biosynthesis of biomass building blocks.

These reaction profiles gave way to the optimal pipeline for system exploration of the resulting solution spaces using Markov chain Monte Carlo sampling [37]. Thus it was possible to establish potential differences in the metabolic states between AP and GP by comparing the allowed specific metabolic solution space. Subsequently, we assessed the most probable carbon flux distribution between the two condition-specific models to reveal integrated information about the predator's metabolism (Fig 6). Thereby the behavior during AP seems to follow a balanced oxidative metabolism aimed at energy production, including intense flux across TCA

and oxidative phosphorylation. On the contrary, no significant fluxes were predicted across anaplerotic and biosynthetic pathways including gluconeogenesis, pentose phosphate, and lipid biosynthesis, which suggests negligible participation of these metabolic hubs during AP. Interestingly, a completely inverse metabolic scenario was predicted under GP. Firstly, this specific model predicted key energetic metabolic pathways being partially inactive during GP. For example, it is important to remark an incomplete performance of the TCA cycle when several stages, including citrate synthase (CS), aconitase (ACONT), isocitrate dehydrogenase (ICDH), and malate dehydrogenase (MDH) were predicted carrying no flux at all. Instead, acetyl-CoA derived from amino acid catabolism was mainly funneled to lipid biosynthesis. Reduction equivalents powering oxidative phosphorylation were produced, almost exclusively, from glutamate metabolism via α-ketoglutarate dehydrogenase and succinate dehydrogenase, thus ensuring ATP production. Finally, a very high flux through gluconeogenesis from pyruvate was predicted, thus enabling the required building blocks for nucleotide and cell envelope biosynthesis in this phase (Fig 6). Interestingly, this scenario described for GP is fully compatible with the energy-saving mechanisms suggested for *B. bacteriovours*. Therefore, the reuse of prey-derived biomass building blocks renders the role of the TCA cycle as the main source of reducing equivalents powering the production of ATP negligible.

## Discussion

Integrative approaches combining traditional and innovative technologies are currently being addressed to establish the metabolic network of hot-spot microorganisms. This issue becomes much more challenging when it refers to predatory microorganisms such as the bacterium *B. bacteriovorus*, which exhibit a bi-phasic lifestyle. With the aim of elucidating the metabolic network wired to predator physiology and lifestyle, we implemented a computational test-bed that proved very useful in the assessment of our predator's phenotype-genotype relationships, while providing new insight on how *B. bacteriovorus*' metabolism operates at the systems level.

### Complex *B. bacteriovorus* lifestyle has guided a significant genome streamlining process and the acquisition of biosynthetic energy-saving mechanisms

Comparison of the essential reactions between *B. bacteriovorus* and other intracellular lifecycle bacteria and free-living microorganisms has revealed the loss of biosynthetic pathways (S3 Table, reactions exclusive to free-living microorganisms). This metabolic scenario is only possible because the host/prey metabolic machinery provides the required biomass building blocks during the intracellular stage of the growth cycle. Despite numerous auxotrophies having been reported in specific genes [6], the metabolic model has allowed the functional contextualization of these biosynthetic deficiencies within the network. For instance, model-based analyses identified additional metabolic gaps which had remained unknown so far, while on the other hand they provided alternative metabolic routes overcoming theoretical auxotrophies. Overall, our analysis has shown a significantly higher number of auxotrophies than previously thought. The loss of essential biosynthetic genes is a typical characteristic of bacteria existing in nutrient-rich environments, such as lactic acid bacteria, endosymbionts or pathogens [76]. In this sense, although *B. bacteriovorus* HD100 possesses a relatively large genome, it could also be included in this "genome streamlining" bacterial group because it directly employs whole molecules from the cytoplasm of the prey [74,77,78]. With regard to the production of the biomass building blocks, it is noteworthy that most amino acids suffer a total lack of biosynthesis pathways. In contrast, *B. bacteriovorus* is fully equipped with the biosynthetic routes for nucleotides and fatty acids. Keeping in mind the macromolecular

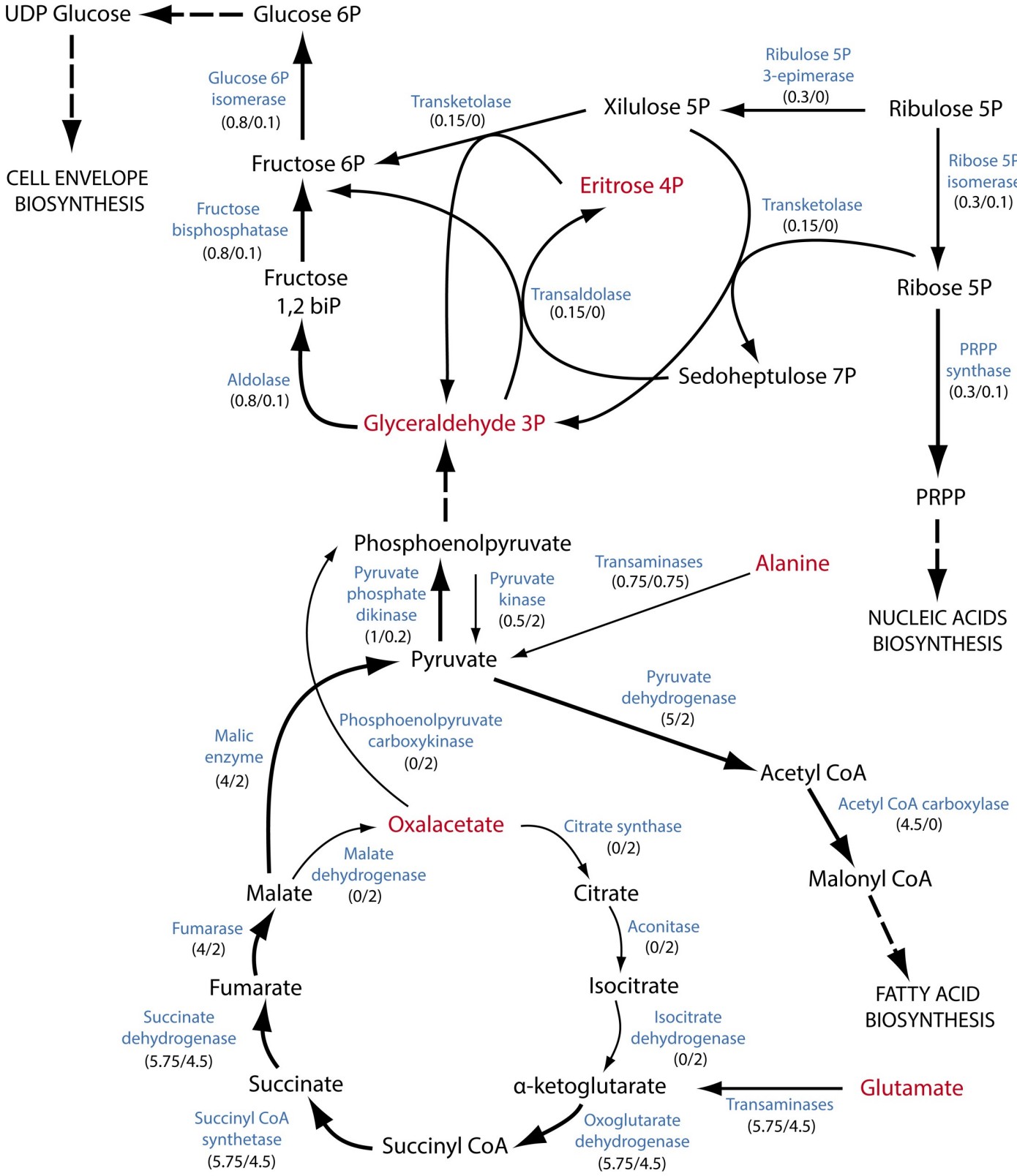

**Fig 6. Prediction of the carbon flux distribution in *i*CH457 metabolic network.** Graphical representation of the metabolic carbon fluxes during the life cycle of *B. bacteriovorus* HD100. The numbers below each reaction represent the more probable flux in each phase (GP flux/AP flux) as determined by Monte Carlo sampling analysis. The thick arrows highlight the carbon flux distribution in GP compared with AP. The thin arrows highlight the reactions that are active in AP compared

with GP. As the major carbon sources are amino acids, alanine and glutamate come directly from the breakdown of the dipeptides or from the single amino acids. Eritrose 4 phosphate and glyceraldehyde 3 phosphate come from the degradation pathways of serine and threonine.

composition of a prokaryotic cell's cytoplasm as the natural growth niche of *B. bacteriovorus* (50% proteins, 20% RNA, 10% lipids, 20% remaining components), it is easy to speculate why the oligopeptide transporter systems are widely represented. While the factors driving *de novo* synthesis or the uptake of biomass building blocks are still unknown, it is likely that these processes are extremely regulated and only activated in the absence of intermediates. A significant flux feeding nucleic acid biosynthesis was predicted (Fig 6). Thus, an important amount of nucleotides came from *de novo* synthesis pathways. This would occur during *in vivo* conditions even in the presence of nucleotides in the extracellular medium (prey´s cytoplasm). This high requirement of nucleotides beyond the amount provided by the prey could justify the presence of a complete nucleotide biosynthesis pathway in contrast with the scenario found in the biosynthesis of amino acids and cofactors when multiple autotrophies were found.

In addition, the presence of these complete metabolic pathways determines the potential ability of the predator to survive and grow without prey, as predicted by the model. Supplying the model with a rich medium based on amino acids returned a simulation which provided key information about growth and generation of biomass. Importantly, this potentially independent growth might be associated with *B. bacteriovorus*' role as a balancer of bacterial population either in aquatic or soil environments, or in the intestine of healthy individuals, because survival of predator cells is not uniquely dependent on the predation event [70].

Related to the essential reactions exclusive of the intracellular microorganism (*Salmonella*, *Shigela*, *Yersinia* and *Bdellovibrio*), it can be highlighted the relevance of the lipid synthesis. These molecules participate in crucial biological processes, including signaling and organization of the membrane of the cells. For intracellular pathogens, it has been described that lipids are also crucial for the interplay with the host cell [79]. The uptake of intracellular pathogens, such as *Salmonella typhimurium* or *Mycobacterium tuberculosis* is led by a re-organization of the lipid microdomains to avoid the degradative environment of the lysosomes. Besides, in concordance with the biphasic life cycle of this intraperplasmic predator, lipid composition determines the structural and functional integrity of the extracellular forms of pathogens [80].

On the whole, our data support the hypothesis and suggest that the metabolic properties of *B. bacteriovorus* are closer to those of the postulated minimal metabolic network. This low robustness of the metabolic network suggests *Bdellovibrio* is more niche-specific than previously thought and the environmental conditions governing predation may be relatively uniform. However, in-depth studies of the metabolic capabilities of the predator are needed to complete the metabolic network and obtain more reliable *in silico* predictions.

## Nutrient availability and biological objective largely conditioned the metabolic shift from *i*CHAP to *i*CHGP

The development of *i*CH457, *i*CHAP and *i*CHGP has provided a computational framework for a better understanding of the physiological and metabolic versatility of BALOs and other predatory bacteria. These models have provided a mechanistic explaining of the required metabolic shift between the different phases. Thus, metabolic fluxes estimations during AP and GP in absence of objective using random sampling are fully compatible with the expected biological objective in these phases e.g., ATP production and growth, respectively. For instance, during GP, several metabolic pathways become inactive, allowing carbon flux distribution re-routing toward biosynthetic pathways. The TCA cycle shifts from a completely operational

state during AP to an anaplerotic mode by inactivating the decarboxilative branch including citrate synthase, aconitase and isocitrate dehydrogenase. In parallel, glutamate was used as a main carbon and energy source. The metabolic switch in *B. bacteriovorus* between the different growth phases has revealed an environmental adaptation of this predator to tackle a rich medium, which would provide an explanation for the development of HI strains. Overall, the carbon flux predictions were compatible with the complex lifestyle of *Bdellovibrio* cells and provided an unprecedented overview of the metabolic shifting required to move from AP to GP, as well as new knowledge about the connections within the predator's metabolic network.

Finally, the results obtained during this study contribute not only to increasing the available metabolic knowledge of *B. bacteriovorus*, but also to providing a computational platform for the full exploitation of this predatory bacterium as a biotechnology workhorse in the near future.

## Supporting information

**S1 Text. Definition of in silico media.**
(DOCX)

**S2 Text. Memote report.**
(HTML)

**S1 Fig. Network analysis of essential reactions.** Essential reactions were grouped associated with the metabolic model. *i*PC815 (green), STMv10 (Pink), *i*JO1366 (brown), *i*CH457 (dark green), *i*AF987 (grey), *i*SF1195 (Orange), *i*JN1411 (blue).
(PNG)

**S1 Table. List of reactions from the final model of *B. bacteriovorus*.**
(XLSX)

**S2 Table. List of metabolites from the final model of *B. bacteriovorus*.**
(XLSX)

**S3 Table. biomass objective function formulation of *i*CH457.**
(XLSX)

**S4 Table. Initial reactions present in the draft and manually curated involved in fatty acid metabolism.**
(XLSX)

**S5 Table. Initial reactions present in the draft and manually curated involved in amino acid metabolism.**
(XLSX)

**S6 Table. Essential reactions extracted from the comparison between the intracellular and the free-living lifestyle.**
(XLSX)

**S7 Table. Specific metabolic content of the models *i*CHAP and *i*CHGP.**
(XLSX)

## Acknowledgments

The authors thank Clive A. Dove for critical reading of the manuscript.

## Author Contributions

**Conceptualization:** M. Auxiliadora Prieto, Juan Nogales.

**Data curation:** Cristina Herencias, Sergio Salgado-Briegas, Juan Nogales.

**Formal analysis:** Cristina Herencias, Sergio Salgado-Briegas, Juan Nogales.

**Funding acquisition:** M. Auxiliadora Prieto, Juan Nogales.

**Investigation:** Cristina Herencias, Juan Nogales.

**Methodology:** Cristina Herencias, Juan Nogales.

**Supervision:** M. Auxiliadora Prieto, Juan Nogales.

**Validation:** Cristina Herencias, Juan Nogales.

**Writing – original draft:** Cristina Herencias.

**Writing – review & editing:** M. Auxiliadora Prieto, Juan Nogales.

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
