## [Decision Letter · Decision Letter 0]

24 Apr 2020

Dear Dr. Nogales,

Thank you very much for submitting your manuscript "Providing new insights on the byphasic lifestyle of the predatory bacterium Bdellovibrio bacteriovorus through genome-scale metabolic modeling." for consideration at PLOS Computational Biology.

As with all papers reviewed by the journal, your manuscript was reviewed by members of the editorial board and by several independent reviewers. In light of the reviews (below this email), we would like to invite the resubmission of a significantly-revised version that takes into account the reviewers' comments.

We cannot make any decision about publication until we have seen the revised manuscript and your response to the reviewers' comments. Your revised manuscript is also likely to be sent to reviewers for further evaluation.

Sincerely,

Vassily Hatzimanikatis

Associate Editor

PLOS Computational Biology

Daniel Beard

Deputy Editor

PLOS Computational Biology

Reviewer's Responses to Questions

**Comments to the Authors:**

Reviewer #1: This manuscript by Herencias et al reports the construction of a metabolic model for the predatory bacterium Bdellovibrio bacteriovorus.

Findings highlight a higher number of amino acid auxotrophies than previously known, while nucleotides and phospholipids can be synthesised de novo by Bdellovibrio.

Experimental validation of some of the model’s predictions (i.e. growth rates in various media) was done using host-independent (HI) mutant of Bdellovibrio, based on the assumption that the mutation leading to host-independency does not affect metabolism. Since the model was based on wild-type genomic and metabolic data, and because it described fairly well the metabolic activity of HI cells, it questions the obligate predatory lifestyle of Bdellovibrio from a metabolic point of view, which is interesting and intriguing.

The authors also constructed metabolic models specific for AP and GP stages by adding in silico constraints to their initial global model (nutrient availability, biological objective, gene expression profile from published RNAseq data).

Major comments:

1) Figure 4: what is the statistical significance of the growth rate differences/similarities?

2) Lines 479-481: from the results section, I had understood that the investment of most resources to ATP generation was an assumption/input (qualified as “biological objective”) for the generation of the iCHAP model (see for ex. lines 366-368), not a conclusion/output from that model. Same remark for next line concerning GP. Please clarify.

3) Figure 5: the authors could discuss what are the intracellular microorganism’s exclusive essential reactions

4) related to the assumption that HI mutants’ metabolism reflects that of the wild-type Bdellovibrio:

- Lines 84-87: can the authors exclude the possibility that the hit mutation has a major impact on metabolism via an indirect regulatory effect?

Minor comments:

- Title: biphasic instead of biphasic

- Line 44: interest in this predatory bacterium, its complex lifestyle (…)

- Line 77: septates instead of septs

- Lines 78-79: whether a large array of hydrolytic enzymes is used to breach the prey OM is not known.

- Line 81-82: it is unclear how HI mimics the dimorphic pattern of HD cells since there is no attack phase stage. Moreover, what is meant by differentiation here is unclear.

- Line 94: septates

- Lines 108-111: this sentence is unclear, please rephrase.

- Line 219: please clarify what such an energy-saving mechanism implies

- Line 224: please clarify how the finding that Bdellovibrio has the ability to synthesise fatty acids and nucleosides de novo, and the reported uptake from the prey, demonstrates a more efficient incorporation of components from the prey.

- Line 249: bacteriovorus

- Line 268: please briefly define what FBA means (for readers who haven’t read the methods section yet).

- Line 305: after constrained, the growth rate

- Line 555: gene/protein

- Figure 1: Model draft instead of mode draft?

- Figure 1: species-specific instead of specie-specific

- S1 text, page 2: major instead of mayor

- S1 text, Table S2: abbreviations; also consider changing “thesis” to “study”.

- Supplementary table .xls file: read me page: present instead of presents

Reviewer #2: This article describes the metabolic reconstruction of the predatory bacterium Bdellovibrio bacteriovorus. The authors test this reconstruction by predicting the bacterium’s growth rate under different growth conditions. The unique nature of this organism makes this reconstruction a useful resource for the scientific community.

While the reconstruction is useful, there isn’t anything novel in the study. There is no novel modeling approach or significant biological insights. Upon reading the abstract and title, I had presumed this study involves a novel predator-prey interaction metabolic model, which would have been very interesting. Instead this study builds a metabolic model using well-established methods and simulates various conditions in silico. The literature validation data provided isn’t extensive either and any new insights reported in the abstract are speculative.

I think the reconstruction process could be greatly improved over its current state. A lot of information about model curation and annotation have not been provided. Finally, a major goal of the reconstruction process is to identify gaps in knowledge and directions to further curate the model. These need to be emphasized in the manuscript.

There is also circular logic in their discussion and results. For their comparison of growth phase (GP) vs attack phase (AP), the authors report that they found no flux in biosynthetic pathway in the attack phase. This is just the consequence of their objective which was maximizing ATP.

Further, they report in the discussion section that they discovered “Through detailed analysis of these specific-condition models it has been possible to conclude that during AP, B.bacteriovorus invests most of its resources in ATP generation… In contrast, GP was characterized by the biosynthesis of biomass building blocks.”

But while performing FBA, they assumed that the AP phase maximizes ATP and the GP maximizes biomass! So it doesn’t make sense to conclude this result based on their assumption. If they had found this using just the transcriptomics data alone then that would have been interesting.

How was reaction directionality determined? was it using literature evidence or other evidence such as basic thermodynamic estimates? This is not discussed in the manuscript

Does the model include growth and non-growth-associated ATP maintenance parameters? if so how were they estimated?

How were Genes-Protein-Reaction annotations done? Was protein complex information taken into account during the reconstruction process to assign GPRs?

Regarding compartments in the model, was some sequence analysis done to determine subcellular location of different enzymes (i.e cytosol or periplasm)?

How many dead end reactions are there in the model? These could be highlighted/discussed for future model curation and to identify less studied parts of the network.

the confidence scores for reactions are not provided in the spreadsheet supplement. The confidence scores are valuable for model curation. The transcriptomic and genomic evidence mentioned earlier can be used to assign these confidence scores (along with known biochemical literature).

How were the reactions mass- and charge balanced?

What other quality controls were performed during the reconstruction process? Did they check for ATP, nadh, nadph production leaks (i.e they should not be produced when there are no inputs)? thiele and palsson, 2010 have a good protocol for testing reconstructions. This can be used as a guideline for reconstructing metabolic networks.

The authors mention that there are gaps in the model. How many gaps were there and how many were filled manually and using automated tools?

The authors report that this reconstruction has high number of transporters and cell wall metabolism enzymes. For example they state “In fact, this category was found to be one of the most representative in terms of number of reactions (161)”. Are these statistics higher than those observed in ecoli or other bacteria? the actual number (161) alone doesnt really convey if transporters are over-represented. This needs to be described in comparison with other organisms.

The authors predicted growth and biomass yield in 5 conditions. They report 100% accuracy in predicting growth in 3 conditions and 70% accuracy in 2 conditions. I think the term accuracy is confusing and misleading in this case. Instead they should say ‘the model predicted 70% of the actual value’.

Please include a pvalue for the rank correlation reported as it is a function of the sample size.

table s2 in the supplementary text: What’s the literature evidence for these reannotations? this should be included in the table

No references or evidence code is provided in s1 table for the curation of these reactions

Other comments

There’s a typo in the title. Biphasic is misspelled.

The abstract uses unnecessary adverbs like ‘unprecedented’, ‘solid’.

Figure 2 would be more informative if it had statistics rather than just words and cartoons describing the process

Notes in s1 table in the spreadsheet are in Spanish

**Have all data underlying the figures and results presented in the manuscript been provided?**

Reviewer #1: No:

Reviewer #2: Yes

PLOS authors have the option to publish the peer review history of their article (what does this mean?). If published, this will include your full peer review and any attached files.

Reviewer #1: No

Reviewer #2: No
---

## [Decision Letter · Decision Letter 1]

20 Jul 2020

Dear Dr. Nogales,

We are pleased to inform you that your manuscript 'Providing new insights on the biphasic lifestyle of the predatory bacterium Bdellovibrio bacteriovorus through genome-scale metabolic modeling.' has been provisionally accepted for publication in PLOS Computational Biology.

Best regards,

Vassily Hatzimanikatis

Associate Editor

PLOS Computational Biology

Daniel Beard

Deputy Editor

PLOS Computational Biology

Reviewer's Responses to Questions

**Comments to the Authors:**

Reviewer #1: The manuscript has been greatly improved by the addition of technical details, explanations, and validation/QC data, allowing a better understanding of the findings from a physiological point of view. The authors provided satisfying answers to most of my concerns and questions.

I just have a few comments and suggestions:

- Line 356/358: "HI strains exhibit a similar lifecycle when growing in a rich medium to the WT strain…" It is unknown if the “lifecycle” is similar: there must be major differences (despite the likely metabolic similarity) between intra-periplasmic and free-living lifestyles; so far the cellular processes at play during HI growth, including elongation, cell division, etc, have not been investigated in sufficient molecular and cellular details to claim similar lifestyles (incl. ref 64 from 1969 and ref 10 from 1977). However I agree with the authors’ response that metabolically speaking, HI strains are supposed to possess similar capabilities. I would try to be more specific in that sentence and avoid firmly stating that HI lifecycle is similar to the WT HD one.

- Legend of Fig 5 is missing

- Line 203: define RPKM

- typos/etc

Line 31: based on

Line 87: multiple fission

Line 132: Gene

Line 133: corresponds

Line 183: represents

Line 199: replace “us”Line 207: taken

Line 286-289: rephrase

Line 294: do you mean “when a specific gene”, instead of “although a specific gene”?

Line 295: Examples of this include

Line 296: it is evident

Line 318: Evolution has

Reviewer #2: The authors have addressed all my concerns

**Have all data underlying the figures and results presented in the manuscript been provided?**

Reviewer #1: Yes

Reviewer #2: Yes

PLOS authors have the option to publish the peer review history of their article (what does this mean?). If published, this will include your full peer review and any attached files.

Reviewer #1: No

Reviewer #2: No

---

## [Editor Report · Acceptance letter]

9 Sep 2020

PCOMPBIOL-D-20-00029R1 

Providing new insights on the biphasic lifestyle of the predatory bacterium Bdellovibrio bacteriovorus through genome-scale metabolic modeling.

Dear Dr Nogales,

I am pleased to inform you that your manuscript has been formally accepted for publication in PLOS Computational Biology. Your manuscript is now with our production department and you will be notified of the publication date in due course.

With kind regards,

Matt Lyles
